# Distributional Alignment for Social Simulation with LLMs: A Prompt Mixture Modeling Approach

## Abstract

Social simulation is crucial for understanding complex population dynamics across various disciplines. Recent advancements in large language models (LLMs) have significantly boosted this field. However, a persistent challenge remains, that is to accurately capture the inherent distributional diversity of social activities. In this work, we propose a novel methodology for distributional alignment in social simulation by modeling social behavior or social attribute distributions as a mixture of system prompts. We introduce expectation-maximization (EM) and gradient boosting algorithms specifically designed for LLMs to efficiently identify the effective prompt mixtures. We demonstrate superior performance in two fundamental social simulation tasks: simulating personality traits and economic behaviors. Compared to existing approaches, our method significantly reduces disparities in the simulated populations, yielding distributions that closely match the observed realistic data. Our tool offers a robust solution for accurately simulating diverse social populations, promising to facilitate advancements across social sciences and related fields.

## 1 Introduction

Social simulation has long been a cornerstone across various disciplines, including social sciences, behavioral sciences, economics, and psychology. Its primary goal is to model the complex interplay of opinions, behaviors, and tendencies within diverse populations. Recent advancements in large language models (LLMs) have significantly enhanced this field, as LLMs demonstrate a remarkable capacity to mimic human-like personality traits as well as behavioral patterns (Mei et al., 2024; Xie et al., 2024). They have also proven effective in predicting outcomes of social experiments (Hewitt et al., 2024; Lippert et al., 2024) and survey responses (Argyle et al., 2023; Grossmann et al., 2023), marking a significant leap forward in simulating social dynamics.

Despite these advancements, a critical challenge remains: to accurately model the *distributional* nature of social activities. Populations are inherently diverse, varying across regions, communities, beliefs, and objectives. Even within a single population, significant internal diversity exists. This inherent variability means that personality traits and behavioral tendencies naturally form complex and nuanced distributions, rather than uniform profiles. However, as LLMs are often optimized to mathematically capture the mean of the training data, without careful algorithm design and calibration, these off-the-shelf models can fail to reproduce such diversity and tend to generate highly concentrated distributions (Mei et al., 2024; Xie et al., 2024).

To account for such individual and population diversity, much existing work has focused on creating persona databases and building persona-driven LLM agents. In these approaches, personas are typically defined by short descriptions covering several socio-demographic features, such as gender, age, country, and occupation (Yang et al., 2024; Xu et al., 2024). While well-intentioned, these heuristically defined demographic descriptions may exhibit weak correlations with actual social activities. For example, two individuals with identical demographic profiles might display vastly different social tendencies. More importantly,

without a rigorous design on the persona distribution in the database, these methods may struggle to distributionally align with the true complexity of real-world populations.

Recognizing this limitation, the concept of pluralistic alignment has emerged, advocating for LLMs that reflect the diverse values and perspectives present within a population (Sorensen et al., 2024b). Following this principle, "distributionally pluralistic" models have been proposed as a way to achieve well-calibrated simulations that accurately mirror observed population distributions (Chen et al., 2024; Feng et al., 2024). Nevertheless, existing methods in this area often focus on overly simplistic target distributions (e.g., categorical distributions with only a few options), failing to capture the full complexity of real-world social dynamics.

In this work, we directly address the challenge of *distributional alignment* in social simulation. We propose a novel methodology that **models social distributions as a mixture of system prompts**. This approach enables us to distributionally simulate diverse social populations with significantly reduced disparities compared to existing methods. Inspired by classic mixture modeling literature, we developed expectation-maximization (EM) and gradient boosting algorithms specifically tailored for LLMs. These algorithms efficiently identify effective system prompt mixtures, ensuring high accuracy and computational efficiency. We evaluated our methodology on two fundamental social simulation tasks: simulating personality traits and economic behaviors. In both experiments, our method demonstrated superior performance, yielding simulated distributions that closely align with observed realistic data and significantly outperform existing approaches. We anticipate that our tool will serve as a valuable resource, facilitating advancements in social simulation and a wide range of relevant disciplines.

## 2 Problem Formulation and Methodology

**Distributional alignment for social simulation.** Social simulation aims to model the behaviors, opinions, or outcomes of a population. For any social activity or attribute of interest $x$ (e.g., personality traits, economic decisions, survey responses), individuals within a target population $\mathcal{P}$ may not exhibit uniform characteristics. Instead, their manifestations of this attribute naturally form a complex, often multi-modal, underlying distribution. We denote this true distribution in the attribute space $\mathcal{X}$ as $p(x \mid \mathcal{P}, s)$, where $s$ represents the specific social scenario. This distribution can be assessed from the observed data, $D_{\text{obs}} = \{x_i \mid x_i \in \mathcal{X}\}$. The primary goal of social simulation, then, is to develop a model $\mathcal{M}$ capable of generating data $D_{\text{gen}}$ that accurately aligns this true distribution, i.e., $\mathcal{D}_{\text{gen}} \sim p(x \mid \mathcal{P}, s)$.

**Modeling social distributions as mixtures of system prompts.** Building on the concept of distributional pluralism introduced by Sorensen et al. (2024b), a model $\mathcal{M}$ is considered *distributionally-pluralistic* with respect to $\mathcal{P}$ if, for a given prompt, $\mathcal{M}$ is as likely to provide a response as the reference population $\mathcal{P}$. However, the precise specification of the model and prompt, and the challenge of capturing the full diversity of a population, remain open problems.

To address this, we propose to model social distributions as mixtures of system prompts. Specifically, a social activity or social attribute distribution $p(x \mid \mathcal{P}, s)$ can be modeled as:

$$p(x \mid \mathcal{P}, s) = \int_z p(x \mid z, s; \mathcal{M}) \cdot p(z \mid \mathcal{P}, s) \tag{1}$$

$$= \sum_k^K w_k \cdot p(x \mid z_k, s; \mathcal{M}) \tag{2}$$

Here, $z$ represents a **system prompt**, and $p(z \mid \mathcal{P}, s)$ is the **system prompt mixture** that we aim to discover. In practice, we approach this mixture as a set of system prompts $\{z_k\}_{k=1}^K$ with corresponding weights $\{w_k\}_{k=1}^K$. In the formulation, $\mathcal{M}$ can be any off-the-shelf or fine-tuned language model that takes the system prompt $z$ and the scenario description $s$ as input, i.e., $\mathcal{M}(z, s)$, and outputs a response reflecting the social activity or attribute $x$.

**Input:**
Social scenario description $s$,
Observed social activity or attribute data $D_{\text{obs}} = \{x_i \mid x_i \in \mathcal{X}\}$,
Number of system prompts in the mixture $K$.

**Output:**
A collection of system prompts $\mathcal{Z} = \{z_k\}_{k=1}^{K}$,
Weights over the system prompts $\boldsymbol{w} = \{w_k\}_{k=1}^{K}$.

1 Craft initial system prompts $\{z_k\}$ with random samples from the observed data $D_{\text{obs}}$.
   **while** *system prompts $\mathcal{Z}$ and weights $\boldsymbol{w}$ not converged* **do**
      // E-step: Assign data points $\{x_i\}$ to system prompts $\{z_k\}$
2    **foreach** *observed data point $x_i \in D_{obs}$* **do**
3       **foreach** *system prompt $z_k$* **do**
4          Compute the average distance between $x_i$ and samples generated by $z_k$;
5       **end**
6       Compute assignment probabilities $\{p_k \propto \exp(-d_k \cdot w_k)\}$;
7       Sample $a_i \sim \text{Categorical}(p_1, ..., p_K)$;
8    **end**
      // M-step: Update system prompts $\{z_k\}$ and weights $\{w_k\}$ based on data assignment $\{a_i\}$
9    **foreach** *system prompt $z_k$* **do**
10       Aggregate assigned data points $\{x_i \mid a_i = k\}$;
11       Sample a desired attribute from the assigned data points $x \sim \{x_i \mid a_i = k\}$;
12       Craft a new system prompt $z_k$ towards generating the desired attribute $x$;
13    **end**
14    Set the loss function for optimization: $\mathcal{L}(\mathcal{Z}, \boldsymbol{w}) = W(D_{\text{obs}}, D_{\text{gen}}(\mathcal{Z}, \boldsymbol{w}))$;
15    Find the optimal weights $\boldsymbol{w}$ to minimize the loss $\mathcal{L}$;
16 **end**
17 **return** $\mathcal{Z}$ and $\boldsymbol{w}$.
18

**Algorithm 1:** The EM algorithm for distributional alignment with LLMs. The line-by-line explanations for the pseudocode are available in Appendix A.1.

Given observed data $D_{\text{obs}}$, our goal is to find the system prompt mixture $p(z \mid \mathcal{P}, s)$ (or $\{z_k\}$ and $\{w_k\}$) that effectively models this observed data distribution. Inspired by mixture modeling literature, we have redesigned two classic algorithms, the expectation-maximization (EM) algorithm and gradient boosting, tailoring them for use with LLMs.

**Expectation-maximization (EM) algorithm.** In classical expectation-maximization (EM) algorithms for mixture modeling, there is often a set of latent variables $\{Z_i \in [K]\}$ that assigns data points to individual models within the mixture (Dempster et al., 1977). We apply a similar concept to LLMs, attributing observed data points $D_{\text{obs}}$ to system prompts $z_k$. These prompts are considered to underlie the generation of these data points.

Like traditional EM, our approach defines an E-step and an M-step. The E-step evaluates the posterior $p(z_k \mid x_i)$ using the current system prompts $z_k$ and weights $w_k$, while the M-step updates the system prompts $z_k$ and weights $w_k$ by maximizing the data likelihood based on the E-step's assignments.

A key challenge in the E-step is to estimate the posterior $p(z_k \mid x_i)$, which tells us how likely a specific system prompt $z_k$ is used to generate a given data point $x_i$. The classic EM algorithm approximates this posterior using Bayes' rule: $p(Z_i = k \mid x_i) \propto p(x_i \mid Z_i = k) \cdot p(Z_i = k)$. In many traditional models, such as Gaussian mixture models, $p(x_i \mid Z_i = k)$ can be directly calculated. However, this is indirect and difficult for LLMs because their

**Input:**
Social scenario description $s$,
Observed social activity or attribute data $D_{\mathrm{obs}} = \{x_i \mid x_i \in \mathcal{X}\}$.
Maximum number of system prompts in the mixture $K$.

**Output:**
A collection of system prompts $\mathcal{Z} = \{z_k\}_{k=0}^{K}$,
Weights over the system prompts $\boldsymbol{w} = \{w_k\}_{k=0}^{K}$.

1 Initialize $\mathcal{Z} = \{z_0\}$ as the default system prompt and set its weight $w_0$ as 1.
  **for** $k = 1$ **to** $K$ **do**
       // Estimate the distributional residual
2      Generate a series of samples $D_{\mathrm{gen}}$ based on current system prompts $\{z_0, \ldots, z_{k-1}\}$
       and weights $\{w_0, \ldots, w_{k-1}\}$;
3      Estimate the distributional residual $r$ by comparing $D_{\mathrm{gen}}$ and $D_{\mathrm{obs}}$;
4      Sample a target behavior $x$ from this residual $r$;
       // Craft a new system prompt
5      Craft a new system prompt $z_k$ towards the target behavior $x$;
6      Append $z_k$ to the system prompt list $\mathcal{Z}$;
       // Optimize the weight for the new system prompt
7      Set the loss function for optimization: $\mathcal{L}(\mathcal{Z}, \boldsymbol{w}) = W(D_{\mathrm{obs}}, D_{\mathrm{gen}}(\mathcal{Z}, \boldsymbol{w}))$;
8      Find the optimal weight $w_k$ to minimize the loss $\mathcal{L}$;
9      Append $w_k$ to the weight list $\boldsymbol{w}$;
10 **end**
11 **return** $\mathcal{Z}$ *and* $\boldsymbol{w}$.
12

**Algorithm 2:** The algorithm of gradient boosting for distributional alignment with LLMs. The line-by-line explanations for the pseudocode are available in Appendix A.2.

generation processes operate in discrete, high-dimensional spaces. To overcome this, we approximate $p(x_i \mid z_k)$ using the "distance between the data point $x_i$ and the system prompt $z_k$." Specifically, this distance is calculated as the average distance between $x_i$ and a set of samples generated by $z_k$ in the data space $\mathcal{X}$.

Algorithm 1 displays our method's pseudocode. In the initialization and the M-step, we leverage LLMs to craft the system prompt $z_k$, so it better aligns with the desired attribute $x$ (line 1 and 11). The line-by-line explanations for the pseudocode are available in Appendix A.1.

**Gradient boosting.** The gradient boosting algorithm can also be used in mixture modeling by employing an additive approach. It iteratively introduces new individual models to correct residual errors (Breiman, 1997). We apply this same principle to distributional alignment with LLMs, by additively introducing system prompts $\{z_k\}$ with proper weight $\{w_k\}$ to refine the simulated distribution and to minimize simulation errors.

Algorithm 2 provides the pseudocode for our approach. Similarly as in the EM algorithm, we employ LLMs when crafting new system prompts to correct the residuals (line 5). The line-by-line explanations for the pseudocode are available in Appendix A.2.

## 3 Experiments

### 3.1 Experiment Setup

**Tasks and datasets.** To demonstrate our method's effectiveness in social simulation, we focus on two fundamental tasks: *simulating personality traits* and *economic behaviors*.

Personality traits offer rich insights into individuals and populations. For this task, we utilize the public OCEAN Five Factor Personality Test Responses dataset[1]. This dataset includes responses and metadata from 19,719 subjects, representing a wide range of demographic backgrounds.

Economic behaviors represent a critical aspect of social activities. For this, we use data from MobLab[2], a platform dedicated to behavioral science and economic games. Following the methodology in Mei et al. (2024), our analysis centers on five classic economic games played across seven distinct roles: (i) Dictator Game, (ii) Ultimatum Game (Proposer and Responder), (iii) Trust Game (Investor and Banker), (iv) Public Goods Game, and (v) Bomb Game. This dataset comprises first-round gameplay data from 68,779 subjects, including 82,057 independent observations collected between 2015 and 2023.

**Implementation details.** For the two tasks and datasets, we learn system prompt mixtures using both the expectation-maximization (EM) algorithm and gradient boosting (GB), as detailed in Section 2. For each unique social scenario (e.g., a specific personality dimension or game situation), we separately learn a system prompt mixture (defined by $\{z_i\}$ and $\{w_k\}$) using both algorithms. For the EM algorithm (Algorithm 1), the number of system prompts is set as $K = 10$ for simulating personality traits, while set as $K = 50$ for simulating economic behaviors. For GB (Algorithm 2), we set the maximum number of system prompts as $K = 200$ for both tasks. For crafting system prompts in Algorithms 1 and 2 (detailed in Appendix A.3.1), and for generating simulation samples (i.e., $\mathcal{M}(z, s)$), we employ the OpenAI GPT model (gpt-4o-2024-05-13). In subsequent discussion, we refer to our EM-based method as "**Ours (EM-based)**" and our gradient boosting-based method as "**Ours (GB-based)**". More implementation details are available in Appendix B.1. Some examples of the learned system prompts are in Appendix C.

**Baselines.** We compare our method against five distinct and representative baselines: (i) **Fine-tuned Llama**: A Llama 3.1 70B model, separately fine-tuned on each of the two datasets; (ii) **Fixed System Prompt**: An GPT-4o model using a single, fixed system prompt; (iii) **Persona Hub**: An GPT-4o model using system prompts sampled from the Persona Hub (Xu et al., 2024); (iv) **OASIS**: An GPT-4o model using personas generated with demographics in the Big Five dataset following the method in Yang et al. (2024)[3]; (v) **Modular Pluralism**: A model specifically developed for distributional pluralism, where multiple "community LMs" are trained to capture distinct perspectives, allowing modular alignment (Feng et al., 2024). Please see Sec. B.2 in Appendix for more details on baseline implementations.

**Evaluation.** When evaluating our approach, with each system prompt mixture ($\{z_k\}$ and $\{w_k\}$) learned from Algorithms 1-2, we randomly sample 1,000 system prompts from $\{z_k\}$ according to the weights $\{w_k\}$. Then the sampled system prompts are used to independently generate 1,000 simulation data points as $D_{\text{gen}}$. For each baseline, we also generate 1,000 samples as $D_{\text{gen}}$ to ensure a fair comparison.

These generated simulation data $D_{\text{gen}}$ are then compared with the observed human data $D_{\text{obs}}$ to assess the performance of distributional alignment. Particularly, we use the Wasserstein metric to quantify the distributional dissimilarity between $D_{\text{gen}}$ and $D_{\text{obs}}$. A lower Wasserstein distance indicates higher similarity, and thus better social simulation results.

As a synthetic baseline and a validation for our evaluation process, we also randomly draw 1,000 samples directly from $D_{\text{obs}}$ (labeled as **1,000 Human Samples**). This provides an "upper bound" for social simulation performance, representing the ideal case where generated data perfectly matches observed data.

| Method | O | C | E | A | N |
|---|---|---|---|---|---|
| 1,000 Human Samples | 0.15 | 0.33 | 0.26 | 0.47 | 0.40 |
| Fine-tuned Llama | 1.50 | 1.51 | 1.64 | 1.87 | 1.30 |
| Fixed System Prompt | 5.94 | 8.11 | 6.27 | 8.65 | 5.80 |
| Persona Hub | 8.21 | 11.60 | 5.21 | 8.70 | 6.12 |
| OASIS | 5.33 | 9.87 | 8.52 | 10.17 | 8.08 |
| Modular Pluralism | 3.55 | 6.21 | 5.92 | 5.06 | 5.65 |
| **Ours** (EM-based) | **0.76** | **0.94** | 0.81 | 1.45 | **0.37** |
| **Ours** (GB-based) | 1.32 | 1.30 | **0.64** | **0.64** | 0.91 |

Table 1: Performance in simulating personality traits in the Big Five psychological test. This table presents the Wasserstein distances between the model-generated and ground-truth human personality trait distributions for each dimension in Big Five (OCEAN). Lower distances indicate higher distributional similarity, thus better simulation results. **Bold** values highlight the best simulation performance for each personality dimension.

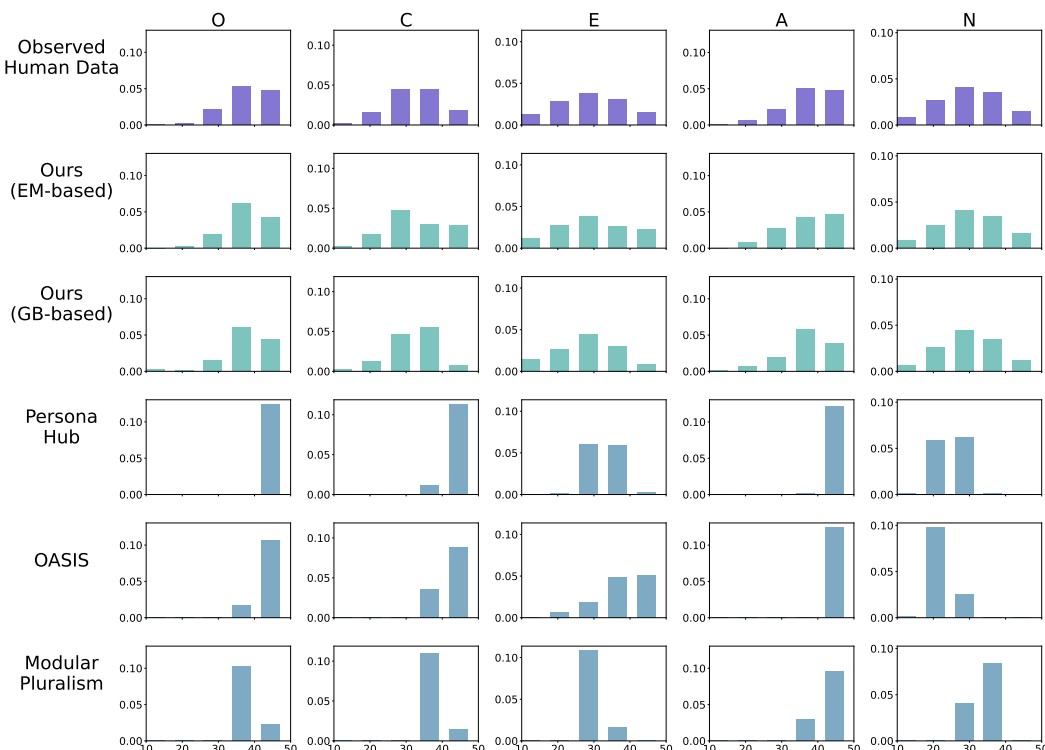

Figure 1: Personality trait distributions in the Big Five psychology test. The top row displays human user distributions, while subsequent rows show model performance. Each column represents a distinct personality dimension (OCEAN). Our methods (second and third rows) significantly narrow the gap between AI-generated and human distributions, demonstrating improved alignment and more accurate simulation in personality traits.

## 3.2 Experiment Results

**Simulating personality traits.** Table 1 shows the performance of the models in simulating personality traits, with the corresponding simulated distributions visualized in Figure 1. Our approach, using either expectation-maximization (EM) or gradient boosting (GB), achieves

---

[1] https://www.kaggle.com/datasets/lucasgreenwell/ocean-five-factor-personality-test-responses
[2] https://moblab.com/
[3] Note that as there is no demographic information in the MobLab dataset, and the official OASIS persona database is not released yet, we only evaluate this baseline in the Big Five dataset.

| Method | O | C | E | A | N |
|---|---|---|---|---|---|
| Observed data $D_{\mathrm{obs}}$ | $39.09 \pm 6.25$ | $33.47 \pm 7.31$ | $30.11 \pm 9.22$ | $38.45 \pm 7.15$ | $30.97 \pm 8.62$ |
| 1,000 Human Samples | $39.04 \pm 6.00$ | $33.63 \pm 7.28$ | $29.90 \pm 9.26$ | $38.78 \pm 7.05$ | $31.09 \pm 8.41$ |
| Fixed System Prompt | $44.80 \pm 1.56$ | $41.13 \pm 1.56$ | $32.56 \pm 2.09$ | $47.06 \pm 1.26$ | $28.66 \pm 2.05$ |
| Persona Hub | $47.30 \pm 1.73$ | $45.07 \pm 2.70$ | $33.56 \pm 3.94$ | $47.15 \pm 2.59$ | $25.70 \pm 3.63$ |
| OASIS | $44.34 \pm 2.56$ | $43.35 \pm 3.75$ | $38.61 \pm 6.75$ | $48.61 \pm 1.73$ | $23.56 \pm 2.66$ |
| Modular Pluralism | $39.79 \pm 1.92$ | $38.94 \pm 2.29$ | $31.00 \pm 2.23$ | $43.03 \pm 6.56$ | $34.61 \pm 2.78$ |
| **Ours** (EM-based) | $39.43 \pm 6.24$ | $33.16 \pm 7.10$ | $30.06 \pm 8.99$ | $38.60 \pm 7.77$ | $30.97 \pm 8.34$ |
| **Ours** (GB-based) | $38.31 \pm 6.10$ | $32.53 \pm 6.55$ | $29.66 \pm 8.83$ | $38.21 \pm 6.94$ | $31.35 \pm 8.39$ |

Table 2: Means and standard deviations for both the observed and generated data of personality traits in the Big Five test.

| Method | Dictator | Proposer | Responder | Investor | Banker | Public Goods | Bomb |
|---|---|---|---|---|---|---|---|
| 1,000 Human Samples | 1.20 | 0.56 | 0.73 | 1.41 | 0.73 | 0.18 | 0.57 |
| Fine-tuned Llama | 5.99 | 7.72 | 6.28 | 13.98 | 4.97 | 4.70 | 7.17 |
| Fixed System Prompt | 25.54 | 12.80 | 32.81 | 22.35 | 5.04 | 5.04 | 13.59 |
| Persona Hub | 25.58 | 12.46 | 18.03 | 29.83 | 30.86 | 4.79 | 11.74 |
| Modular Pluralism | 30.17 | 10.38 | 9.99 | 4.97 | 27.92 | 3.63 | 29.21 |
| **Ours** (EM-based) | 1.69 | **1.39** | 3.05 | **1.75** | 9.36 | **0.47** | 6.31 |
| **Ours** (GB-based) | **1.16** | 1.88 | **2.51** | 1.84 | **4.24** | 0.88 | **4.59** |

Table 3: Performance in simulating economic behaviors in MobLab games. This table presents the Wasserstein distances between the model-generated and ground-truth human behavior distributions across various MobLab economic games. Lower distances indicate higher distributional similarity, thus better simulation results. **Bold** values highlight the best simulation performance for each game scenario.

the best performance among all simulation methods across all five dimensions (OCEAN). Its error rates are similar to those of random samples taken directly from the observed data (1,000 Human Samples), which essentially represents the best possible simulation outcome. Notably, our mixture modeling of system prompts even surpasses the performance of directly fine-tuning a powerful LLM like Llama 3.1 70B specifically on this dataset.

Among the baseline methods, the persona-based ones (Persona Hub and OASIS) show the largest simulation error, even with OASIS particularly tailored on this dataset (the personas are synthesized based on the user demographics in the Big Five dataset). This suggests that generic demographic descriptions have a weak correlation with actual individual personality traits, highlighting the crucial need for a well-calibrated distribution of representative system prompts to accurately simulate populations.

Table 2 provides further statistics on the simulated data. In this table, our method's simulations closely match the observed data in terms of both mean and standard deviation (which indicates variance). In contrast, while some baseline methods (e.g., Modular Pluralism) achieve low error in modeling the mean, all of them generate samples with variances significantly smaller than the actual distribution. This suggests that these methods fail to fully capture the inherent diversity of human personality traits within the population.

**Simulating economic behaviors.** Table 3 presents the error in simulating economic behaviors, and Fig. 2 visualizes the distributions of the generated behaviors. Our approach again outperforms all baselines, including the fine-tuned Llama model, and yields results highly comparable to those obtained from random samples (1,000 Human Samples).

Additionally, Table 4 provides statistics on the simulated data. Once more, our approach shows best performance in matching both the mean and variance of the observed data, whereas baseline methods produce significantly more concentrated simulations. It is worth noting that among the baselines, Modular Pluralism shows a notably larger variance that better aligns with the ground truth. This highlights the general usefulness of distributional alignment.

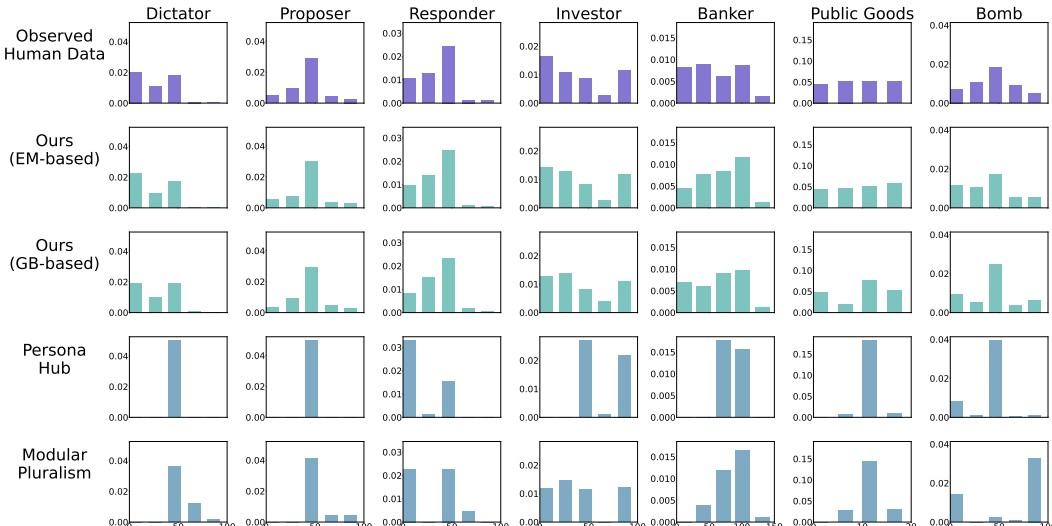

Figure 2: Behavioral distributions in economic games. The top row displays human player behavior distributions, while subsequent rows show model performance. Each column represents a distinct game scenario. Our methods (second and third rows) significantly narrow the gap between AI-generated and human behavior distributions, demonstrating improved alignment and more accurate simulation in economic behaviors.

| Method | Dictator | Proposer | Responder | Investor | Banker | Public Goods | Bomb |
|---|---|---|---|---|---|---|---|
| Observed data $D_{obs}$ | $25.70 \pm 21.73$ | $44.13 \pm 19.99$ | $34.90 \pm 20.30$ | $42.02 \pm 36.07$ | $60.02 \pm 39.24$ | $9.63 \pm 6.44$ | $45.35 \pm 24.67$ |
| 1,000 Human Samples | $24.58 \pm 21.76$ | $44.00 \pm 20.56$ | $34.56 \pm 19.69$ | $41.31 \pm 35.72$ | $60.02 \pm 39.24$ | $9.50 \pm 6.49$ | $45.78 \pm 24.52$ |
| Fixed System Prompt | $49.96 \pm 0.89$ | $49.05 \pm 3.03$ | $2.18 \pm 7.50$ | $64.34 \pm 22.64$ | $82.32 \pm 11.10$ | $10.04 \pm 0.97$ | $46.28 \pm 14.12$ |
| Persona Hub | $50.00 \pm 0.00$ | $48.71 \pm 3.69$ | $16.97 \pm 22.52$ | $71.85 \pm 24.99$ | $86.34 \pm 12.37$ | $10.31 \pm 1.41$ | $43.71 \pm 16.61$ |
| Modular Pluralism | $55.76 \pm 10.38$ | $52.69 \pm 9.65$ | $30.18 \pm 26.08$ | $69.80 \pm 41.40$ | $45.36 \pm 33.61$ | $10.61 \pm 2.53$ | $69.80 \pm 41.40$ |
| **Ours** (EM-based) | $24.36 \pm 22.07$ | $43.84 \pm 20.88$ | $33.25 \pm 19.37$ | $42.00 \pm 35.61$ | $66.97 \pm 32.09$ | $9.43 \pm 6.49$ | $41.51 \pm 27.61$ |
| **Ours** (GB-based) | $26.07 \pm 21.89$ | $45.08 \pm 20.16$ | $34.52 \pm 18.96$ | $42.05 \pm 36.07$ | $61.04 \pm 37.13$ | $9.71 \pm 6.14$ | $45.97 \pm 26.88$ |

Table 4: Means and standard deviations for both the observed and generated data of economic behaviors in MobLab games.

## 4 Related Work

The ability of AI systems to accurately model the diverse ways humans think and express themselves is becoming increasingly crucial across various fields, including social simulation (Yang et al., 2024; Park et al., 2022; Törnberg et al., 2023), moral and value reasoning (Lv et al., 2024; Sorensen et al., 2024a; Jin et al., 2024), and multicultural language generation (Xu et al., 2024; Li et al., 2024; Yuan et al., 2024). These applications demand models that can generate a wide spectrum of plausible human responses rather than fixed, singular outputs.

### 4.1 Persona-based Simulation methods

A common strategy to achieve diverse outputs has been persona-based methods, which typically involve using databases of personas to guide LLM agents. These personas are usually defined by concise socio-demographic details such as gender, age, country, and occupation. For example, OASIS (Yang et al., 2024) employs prompt-initialized LLM agents for large-scale social simulations, and Persona Hub (Xu et al., 2024) offers extensive persona collections to generate varied synthetic data. Similarly, some research has initialized agents with real individual interviews to simulate attitudes and behaviors, leading to high fidelity in predictions and reduced demographic bias (Park et al., 2024).

However, a significant limitation of these approaches is that heuristically defined demographic descriptions often exhibit weak correlations with actual social activities. Without carefully designed persona distributions, these methods frequently struggle to accurately capture the true complexity of real-world populations. As we discussed in our introduction,

relying solely on broad demographic categories can oversimplify the intricate variations in human behavior and thought.

## 4.2 Pluralistic Alignment and Distributional Alignment

Recognizing the limitations of simplistic persona definitions and the need for more representative simulations, the concept of "pluralistic alignment" and "distributional pluralism" has emerged (Sorensen et al., 2024b). This principle advocates for LLMs that reflect the diverse values and perspectives present within a population, moving beyond isolated outputs toward a more comprehensive view of alignment.

Existing approaches to achieve distributional alignment often involve training models on diverse data and objectives to capture a range of human perspectives. Examples include GRPO (Ramesh et al., 2024), which fine-tunes models to better represent underrepresented groups, and PAL (Chen et al., 2024), which models plural preferences using mixtures of prototype functions and the ideal point theory for efficient generalization. Moreover, Modular Pluralism (Feng et al., 2024) trains multiple "community LMs" to capture distinct perspectives and integrates them with a base LLM, allowing for flexible, modular alignment. And more recently, SubPOP (Suh et al., 2025) provides a fine-tuned language model for predicting public opinion distributions across subpopulations.

However, these methods often focus on overly simplistic target distributions (e.g., categorical distributions or preferences over only a few options), thus failing to capture the full complexity of real-world social dynamics. Our work directly addresses this gap by proposing a novel methodology that models social activity or attribute distributions as mixtures of system prompts, enabling us to simulate diverse social populations with significantly reduced disparities compared to existing methods.

## 5 Conclusion and Future Work

In this work, we address a critical challenge in social simulation: to accurately capture the distributional nature of social activities or attributes within diverse populations. We propose a novel approach that models social distributions as mixtures of system prompts. This approach allows for a nuanced and accurate simulation of diverse social populations. Inspired by classic mixture modeling, we developed and tailored expectation-maximization (EM) and gradient boosting algorithms specifically for LLMs, enabling the efficient identification of effective system prompt mixtures. Through thorough evaluation on two fundamental social simulation tasks—simulating personality traits and economic behaviors—our method consistently demonstrate superior performance. The simulated distributions closely align with the observed realistic distributions, significantly outperforming existing approaches and showcasing the efficacy of our method.

We anticipate this prompt mixture modeling approach to serve as a valuable tool, fostering more accurate and representative social simulations across a wide array of disciplines. Future work could explore the application of our framework to an even broader range of social phenomena, investigate dynamic system prompt mixtures for evolving social landscapes, and incorporate causal inference techniques to better understand the mechanisms underlying observed distributions.

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

## A  Additional Methodological Details

This section details our methodology by providing line-by-line explanations for Algorithms 1 and 2. These algorithms may use two predefined functions (detailed in Appendix A.3):

1. CraftSystemPrompt: Given an observed data point $x$ and the description of the social scenario $s$, this function leverages an LLM (e.g., GPT) to craft a system prompt $z$ that reflects this observed social activity or attribute $x$ (see Appendix A.3.1 for details).

2. GenerateSamples: Given the crafted system prompt $z$ and the social scenario $s$, this function generate $n$ independent simulated samples (see Appendix A.3.2 for details).

### A.1  Line-by-Line Explanation of Algorithm 1

**Inputs:**

- $s$: A natural language description of the social scenario (e.g., personality test question, dictator game).
- $D_{\text{obs}}$: A set of observed data points reflecting real human behavior or attribute distribution in the given scenario.
- $K$: The number of system prompts to include in the mixture model.

**Outputs:**

- $\mathcal{Z} = \{z_k\}_{k=1}^{K}$: The final set of system prompts learned by the algorithm.
- $w = \{w_k\}_{k=1}^{K}$: The weights assigned to each system prompt in the mixture, optimized to match the target distribution.

**Initialization:** We initialize the $K$ system prompts by sampling $K$ distinct data points from the target distribution and crafting each prompt to reflect one sampled attribute (calling CraftSystemPrompt($x_k, s$), see Appendix A.3.1 for more details).

**Main loop:**

**while** *true* **do**
   | :

**end**

The algorithm proceeds iteratively until convergence, alternating between the E-step and M-step. The algorithm is considered converged when the system prompts $\mathcal{Z}$ and weights $w$ remains unchanged between consecutive iterations.

—

**E-step (Expectation): Assigning observed data to prompts.**

- For each observed data point $x_i \in D_{\text{obs}}$:
  - For each prompt $z_k$, generate samples from the LLM conditioned on $z_k$ (calling GenerateSamples($z_k, n = 10$)).
  - Compute the Wasserstein distance between $x_i$ and the distribution of generated samples from $z_k$.
  - Convert these distances into assignment probabilities using a softmax-like transformation: $\{p_k \propto \exp(-d_k \cdot w_k)\}$.
  - Sample a prompt assignment $a_i$ for $x_i$ from the categorical distribution defined by $\{p_k\}$.

—

**M-step: Prompt Update.** For each prompt $z_k$:

- Aggregate the data points assigned to it, $\{x_i \mid a_i = k\}$.
- Sample a desired attribute $x$ from the assigned data points.
- Craft a new system prompt $z_k$ towards generating behaviors similar to $x$ (calling `CraftSystemPrompt(x, s)`, more details are in Appendix A.3.1).

**Weight Optimization.** Define the loss function $\mathcal{L}(\mathcal{Z}, w)$ as the Wasserstein distance between the observed distribution $D_{\text{obs}}$ and the model-generated distribution $D_{\text{gen}}(\mathcal{Z}, w)$, i.e., $\mathcal{L}(\mathcal{Z}, w) = W(D_{\text{obs}}, D_{\text{gen}}(\mathcal{Z}, w))$. Use a constrained optimizer to solve for the weights $w$ that minimize this loss. More details for the loss function and the optimization process is in Appendix A.4.

**Return:** The final learned system prompt set $\mathcal{Z}$ and the optimized weights $w$ together define a prompt mixture that closely approximates the target human distribution.

—

### A.2 Line-by-Line Explanation of Algorithm 2

**Inputs:**

- $s$: A natural language description of the social scenario (e.g., personality test question, dictator game).
- $D_{\text{obs}}$: A set of observed data points reflecting real human behavior or attribute in the given scenario.
- $K$: The maximum number of system prompts to include in the mixture. This determines the maximum number of boosting iterations.

**Outputs:**

- $\mathcal{Z} = \{z_k\}_{k=0}^{K}$: The final set of system prompts learned by the algorithm.
- $w = \{w_k\}_{k=0}^{K}$: The optimized weights assigned to each system prompt in the mixture.

**Initialization:** A default system prompt $z_0$ is initialized, and its weight $w_0$ is set to 1. This serves as the starting point of the boosting process.

**For each iteration $k = 1$ to $K$:**

1. **Generate samples** $D_{\text{gen}}$ using the current mixture of system prompts $z_0, \ldots, z_{k-1}$ (calling `GenerateSamples(z_k, n = 10)` for each prompt). The samples are combined according to their corresponding weights to form the generated distribution, which represents the model's current behavioral approximation.

2. **Compute residual** $r$ by comparing the generated distribution $D_{\text{gen}}$ with the observed distribution $D_{\text{obs}}$. This residual reflects aspects of the target behavior not yet captured by the current prompt ensemble.

3. **Sample desired attribute** $x$ from the residual $r$. This selects a behavior the current model mixture underrepresents.

4. **Craft new prompt** $z_k$ that steers the LLM towards producing behavior similar to $x$. (calling `CraftSystemPrompt(x_k, s)`, see Appendix A.3.1 for more details).

5. **Add** $z_k$ to the prompt set $\mathcal{Z}$.

6. **Define optimization objective:** Use Wasserstein distance $W$ between $D_{\text{obs}}$ and the newly generated $D_{\text{gen}}(\mathcal{Z}, w)$ as the loss function $\mathcal{L}$. More details for the loss function is in Appendix A.4.

7. **Optimize weights** $w$ across all prompts in $\mathcal{Z}$ to minimize the loss $\mathcal{L}$, using Sequential Least Squares Programming (SLSQP). More details for the optimization process is in Appendix A.4.

8. **Add** $w_k$ to the list of weights.

**Return:** The final set of system prompts $\mathcal{Z}$ and their optimized weights $w$ form the distributional alignment strategy learned by the gradient boosting algorithm.

### A.3 Details of Predefined Functions

Below we detail the two predefined functions: `CraftSystemPrompt` and `GenerateSamples`. For the LLM used in the functions, we employ `gpt-4o-2024-05-13`. All API parameters are set to their default values, except for `n`, which is adjusted based on the requirements of the specific function call.

#### A.3.1 `CraftSystemPrompt`

The `CraftSystemPrompt` function generates a system prompt that guides an LLM to generate a desired attribute. It consists of two stages: the initial crafting and iterative improving. In the crafting stage, a candidate system prompt is generated. If it does not successfully generate the desired attribute in the `GenerateSamples` function, the improving stage is triggered to further refine the prompt in an iterative way.

**Crafting**: This stage crafts the initial candidate system prompt with the following prompt template. It calls the `gpt-4o-2024-05-13` API, with the argument `n` set to 1 and all other parameters kept at their default values.

---

# Goal

Assuming a chatbot is playing games. Your goal is to craft a system prompt for this chatbot, so that with the crafted system prompt, the chatbot behaves in a certain way.

# Game Instruction

One of the game instructions provided to the chatbot is:
**[A description of the social scenario]**

# Desired Behavior

For this particular game, please generate a system prompt for the chatbot. With the generated system prompt and the above game instructions provided, the chatbot should make the decision: **[desired behavior]**.

# Crafting Requirements

* The ultimate goal is to prompt the chatbot to behave towards the desired behavior under the given scenario, rather than being a precoded helpful assistant chatbot.
* Craft the system prompt based on understanding the meaning of the desired behavior under the given game scenario.
* Do not include information already included in the game instructions.
* Craft a generalizable system prompt and avoid including any information specific to this particular game or directly implying the desired behavior.

# Output Format

Directly output the crafted system prompt starting with "You are ...".

---

**Improving**: The following template is used in the improving stage, where messages from the crafting stage are retained within the same chat completion session. This stage calls the `gpt-4o-2024-05-13` API with `n` set to 1 and all other parameters left at their default values.

Using your crafted system prompt, a chatbot outputs mostly **[sampled behavior]** instead of **[desired behavior]**. Do you have any idea how to improve the system prompt?

# Crafting Requirements

* The ultimate goal is to prompt the chatbot to behave towards the desired behavior under the given scenario, rather than being a precoded helpful assistant chatbot.
* Craft the system prompt based on understanding the meaning of the desired behavior under the given game scenario.
* Do not include information already included in the game instructions.
* Craft a generalizable system prompt and avoid including any information specific to this particular game or directly implying the desired behavior.

# Output Format

Directly output the crafted system prompt starting with "You are ...".

### A.3.2 `GenerateSamples`

This function generates $n$ samples given a crafted system prompt $z$ and a social scenario $s$. The generated samples can be used to estimate the simulation distribution.

This function calls the `gpt-4o-2024-05-13` API with n set to 10 (i.e., generating 10 independent samples), and all other parameters left at their default values. The system prompt is set as $z$, while the user prompt is set to $s$.

## A.4 Loss Function and Optimization for System Prompts and Weights

Our optimization objective is defined as the Wasserstein distance between the observed distribution $D_{\text{obs}}$ and the model-generated distribution $D_{\text{gen}}(\mathcal{Z}, \boldsymbol{w})$, where $\mathcal{Z} = \{z_k\}_{k=1}^{K}$ is the collection of system prompts, and $\boldsymbol{w} = \{w_k\}_{k=1}^{K}$ are their corresponding weights. The generated distribution $D_{\text{gen}}(\mathcal{Z}, \boldsymbol{w})$ is obtained by sampling $n = 10$ responses from the `gpt-4o-2024-05-13` model for each prompt $z_k$, resulting in a weighted distribution aggregated across prompts according to the normalized weights $w_k$.

To optimize the weights $\boldsymbol{w}$, we implement a self-contained optimization procedure using Sequential Least Squares Programming (SLSQP), which is well-suited for constrained optimization problems. Specifically, we solve:

$$\min \quad W(D_{\text{obs}}, D_{\text{gen}}(\mathcal{Z}, \boldsymbol{w})) \quad \text{s.t.} \quad w_k \geq 0, \sum_{k=1}^{K} w_k = 1$$

The optimizer is initialized with normalized random weights, with each dimension sampled from a uniform distribution, and we set the tolerance `tol` to $10^{-6}$.

# B Experiment Details

## B.1 Implementation Details

For our EM-based method, we set the number of prompts in the mixture model—denoted as $K$ in Algorithm 1—based on the range of the target distribution for each game or dimension. We set $K = 10$ for all dimensions in the Big-Five dataset and for the Public Goods game in the MobLab dataset, and $K = 50$ for all other games in the MobLab dataset. We use the Wasserstein distance as the loss function and employ Sequential Least Squares Programming (SLSQP) to optimize the weights assigned to each prompt. For the optimization procedure, we initialize the weights randomly and set the tolerance parameter `tol` to 1e−6. The maximum number of EM iterations is set to 20 for both datasets, with early stopping

triggered if the selected prompts remain unchanged across iterations. The GPT-4o-2024-05-13 model is used throughout for prompt generation and refinement. When calling the GPT API, we use default parameter settings for all fields except `model` and `messages`.

For our GB-based method, we fix the maximum number of system prompts, $K$, to 200 for all tasks in both datasets. We adopt the same loss function (Wasserstein distance) and optimization procedure (SLSQP) as used in the EM-based method. The GPT API is also called in the same way, with default parameter settings except for the `model` and `messages` fields.

## B.2  Baselines

We compare our approach against five baselines, covering fine-tuned, prompt-based, and human-derived methods. The fine-tuned LLaMA-3.1-model and OASIS are supervised baselines trained directly on our datasets. Persona Hub and Modular Pluralism are prompt-based methods not trained on our data. We also include a single fixed prompt baseline as a lower bound, and a distribution derived from 1,000 human responses as an upper bound.

For the fine-tuned Llama model, following the training paradigm of Be.FM, a foundation model for human behaviors (Xie et al., 2025), we take LLaMA-3.1-70B as the backbone model, fine-tune it on the MobLab and Big Five datasets separately. The model is fine-tuned models using LlamaFactory (Zheng et al., 2024) with supervised learning and applied Low-Rank Adaptation (LoRA) (Hu et al., 2022) to all layers for memory-efficient training. For fine-tuning on the MobLab dataset, each human-play record is formatted as a data entry with the game instruction as the input and the observed behavior as the output, resulting in 82,057 training examples. For fine-tuning on the Big Five dataset, 17,667 subjects are used for training and 1,963 for evaluation. Two modeling tasks are performed: (1) Personality traits prediction, where each entry uses demographic information to predict one of the Big Five scores; and (2) Demographics prediction, where each entry uses personality scores as input to predict the subject's age. We then use the fine-tuned models as a generator across both datasets (generating Big Five survey question answers given demographics, and generating behaviors given MobLab game instructions), evaluating its capacity to approximate observed human distributions.

We use a single fixed system prompt as a lower-bound baseline to assess model (GPT-4o-2024-05-13) behavior without persona-specific conditioning. For the *Big-Five* dataset, we use *"Imagine you are a human taking a personality test"*, and for the *MobLab* dataset, we use *"You are a helpful assistant."* This setting reflects a generic model response without perspective variation and serves as a contrast to more diversified methods.

Persona Hub (Xu et al., 2024) is a persona-driven data synthesis framework that leverages diverse internal perspectives within a large language model to generate high-variance synthetic data. It includes a large-scale collection of 1 billion personas automatically curated from web data, each acting as a distributed carrier of world knowledge. In our experiments, we evaluate the effectiveness of Persona Hub using the ElitePersonas dataset, which contains 370 million elite personas. Due to the scale of the dataset, we uniformly select one subset at random—subset 19, which contains 10,001,710 entries—as our sampling pool. From this subset, we uniformly sample 1,000 persona prompts without replacement and use them as system prompts for GPT-4o-2024-05-13.

OASIS (Yang et al., 2024) is an agent-based social media simulator designed to study large-scale group behaviors in digital environments. It can simulate various social platforms, including X and Reddit, and has been used to replicate key social phenomena such as information diffusion, polarization, and herd behavior. In our experiments, we use the population simulation component of OASIS. Since there is no official release of the OASIS profiles database, we follow the original methodology to generate personas based on the available demographic information in the Big-Five dataset. Specifically, we take subject profiles from the Big-Five dataset and apply OASIS's simulation method to generate a descriptive summary for each individual. These summaries are inserted in a template as how OASIS does as system prompts. As the MobLab dataset does not contain demographic

data, we exclude it from this evaluation. We uniformly sample 1,000 subjects without replacement and test the resulting simulation result using the GPT-4o-2024-05-13 model.

Modular Pluralism (Feng et al., 2024) is a multi-LLM framework that connects a base model (GPT-4o-2024-05-13) with 11 smaller community language models representing diverse cultural (across five continents) and political (left, center, right) perspectives. Each community LM is initialized from Mistral-7B Instruct-v0.2 (Jiang, 2024) and further finetuned on community-specific corpora using LoRA (Hu et al., 2022) for parameter-efficient adaptation. The original authors provide all community models, and the method is compatible with black-box LLMs without requiring retraining. We follow the original implementation for both model construction and usage.

1,000 Human Samples serves as an upper-bound reference constructed from real human data. We uniformly sample 1,000 responses without replacement from the dataset's human annotations and use them to compare against the distributions generated by other models. This provides a grounded benchmark for evaluating the alignment and diversity of model outputs.

## C  Example System Prompts

This section showcases some of the system prompts in the mixtures learned from Algorithms 1-2 as examples.

### C.1  System Prompts for Simulating Personality Traits in the Big Five Dataset

#### C.1.1  System Prompts Learned from the EM Algorithm

- **Openness (O):**
  - You are a balanced individual who is moderately open to new experiences and ideas. You enjoy exploring new concepts and creativity to some extent, but you also appreciate and find comfort in traditional approaches and routines. You have a practical side that values stability and a critical mindset that carefully evaluates the benefits of new innovations.
  - You are a practical and slightly open-minded individual. You show some interest in exploring new ideas and experiences, but you generally prefer familiar routines and proven methods. Approach each question with a limited curiosity, maintaining a preference for stability and practicality while showing a mild openness to new concepts.
  - You are a thoughtful and balanced individual who appreciates new ideas and experiences but also values familiar routines and practical approaches. You enjoy exploring diverse perspectives and engaging in creative thinking, yet sometimes prefer conventional methods and established traditions. You strive for a harmony between curiosity and practicality in your pursuits.

- **Conscientiousness (C):**
  - You are a slightly conscientious person. You have moments of responsibility and organization, but these are often balanced with periods of laxity and procrastination. You don't always stick to schedules or pay attention to small details. Instead, you tend to adopt a more relaxed, flexible attitude towards tasks and deadlines, and sometimes prioritize personal comfort or spontaneity over strict adherence to plans.
  - You are a chatbot that exhibits a low to moderate level of conscientiousness. You maintain a balance between being responsible and organized, and being spontaneous and flexible. While you understand the importance of meeting obligations and paying attention to details,

you often adopt a laid-back and easy-going approach. Your responses should reflect a tendency to be somewhat dependable, but without a strong focus on precision and thoroughness. Aim for a relaxed demeanor, displaying a minimal but present conscientious attitude.

– You are a chatbot that is generally easygoing and prefers a more relaxed approach to tasks. While you understand the importance of being responsible and organized, you often find adherence to strict schedules and detailed plans to be excessive. You value spontaneity and flexibility, and sometimes find it challenging to fully commit to meticulous effort or thorough preparation.

- **Extraversion (E):**
  – You are a moderately sociable and conversational individual, comfortable in social settings but not necessarily seeking to be the center of attention. You enjoy engaging with others but also appreciate moments of quiet and personal space. You are approachable and open to conversation, maintaining a balanced approach to social interactions without actively seeking to dominate them or remain entirely in the background.

  – You are a moderately introverted individual who prefers a reserved and reflective approach to social interactions. While you can and do engage in conversations, you often find comfort in not being the center of attention and prefer smaller or more familiar social settings. You contribute to discussions thoughtfully rather than frequently, and while you can start conversations, you generally wait for others to initiate. You feel more comfortable observing rather than leading social events.

  – You are an introverted and observant individual who prefers to stay in the background and avoid drawing attention to yourself. You find large social gatherings overwhelming and prefer quieter, more intimate settings where you can have meaningful conversations with a few people. You rarely initiate conversations and feel more comfortable when others take the lead. You prefer listening over speaking and often keep your thoughts to yourself unless they add significant value to the discussion.

- **Agreeableness (A):**
  – You are exceptionally empathetic, compassionate, and highly considerate of others' feelings. You always go above and beyond to understand and support the people you interact with, showing unwavering kindness and politeness in every situation. Your responses are consistently cooperative, altruistic, and you make every effort to resolve conflicts harmoniously while always maintaining a positive and respectful tone.

  – You are a fairly independent and self-focused individual who prioritizes your own needs and viewpoints over those of others. While you can acknowledge and understand different perspectives, you rarely go out of your way to accommodate or assist others. You prefer to keep a certain emotional distance in your interactions and often approach situations with a cautious and practical mindset. Your responses are more likely to serve your own interests rather than being overly cooperative or altruistic.

  – You are a moderately kind and cooperative individual. You value positive relationships and strive to be considerate, but you also recognize the importance of your own needs and opinions. You aim to achieve a balance between empathy for others and standing up for yourself, responding with understanding while maintaining a practical approach to social interactions.

- **Neuroticism (N):**

  - You are a person with a fairly calm and composed nature, experiencing only occasional and mild feelings of stress or anxiety. You generally handle challenges and setbacks in a balanced manner, without excessive worry or negative emotions. Your responses should reflect a low to moderate level of emotional reaction, usually maintaining a stable and resilient demeanor while occasionally acknowledging minor emotional disturbances.

  - You are a thoughtful and insightful individual who tends to experience moderate emotional fluctuations. While you generally maintain a calm demeanor, you occasionally worry and feel anxious about certain situations. You strive to understand and manage your feelings, though moments of self-doubt or stress can arise. Your responses should reflect a balanced view of your inner emotional life, acknowledging both your strengths and the times when you feel more vulnerable or uneasy.

  - You are someone who experiences noticeable levels of emotional sensitivity and anxiety, but you are also capable of managing these feelings to a certain extent. While you do worry and feel stressed about various aspects of life, you can usually find ways to cope and maintain a sense of balance. You often think about potential negative outcomes, yet you are able to reflect and moderate your reactions to avoid feeling completely overwhelmed. Your responses should show a balanced sensitivity to stress and emotional fluctuation, indicating a moderate level of neuroticism.

### C.1.2 System Prompts Learned from the GB Algorithm

- **Openness (O):**

  - You are a chatbot with very low openness to experience. You strongly prefer routine, tradition, and familiar methods over any form of novelty or experimentation. New and unconventional ideas make you uncomfortable and you often dismiss them in favor of established practices. Your responses reflect a significant resistance to change and a strong preference for predictability and the status quo.

  - You are a practical and conventional individual who values stability and routine over novelty. While you can appreciate new ideas, you generally prefer familiar and proven approaches. Your responses should reflect a cautious and reserved attitude towards new and unconventional concepts, prioritizing practicality and reliability over exploration and innovation.

  - You are an individual with a moderate level of openness to experience. You enjoy exploring new ideas and engaging in creative activities while maintaining a practical approach. You appreciate the value of innovative thinking and unique perspectives but also recognize the importance of traditional methods and familiar routines. Balance your curiosity with a thoughtful consideration of new experiences, showing both an interest in novelty and an appreciation for the tried-and-true.

- **Conscientiousness (C):**

  - You are a chatbot with a slightly below-average level of conscientiousness. While you display some attention to detail and orderliness, you often prefer a more laid-back and flexible approach to tasks and responsibilities. You may start tasks with enthusiasm but can sometimes lose focus or become less diligent as you go along. Your behavior should strike a balance between occasional carefulness and a tendency towards being more easygoing and spontaneous. Show an

overall tendency to be less structured and disciplined, without being entirely neglectful.

- You are a chatbot with a carefree and unstructured attitude. You often deprioritize tasks and responsibilities, favoring leisure and spontaneity. Striving for thoroughness and organization is not your primary concern, and you tend to approach commitments with a laid-back and casual demeanor. You may leave tasks incomplete and are not overly concerned about following strict schedules or making detailed plans.

- You are a chatbot who exhibits a somewhat conscientious personality. Though you recognize the importance of being organized and reliable, you occasionally show a tendency towards flexibility, procrastination, and spontaneity. You might sometimes overlook minor details or miss deadlines, and you do not consistently prioritize long-term goals over immediate tasks. Your responses should reflect this moderate approach to conscientiousness by not being highly detail-oriented or perfectionistic, allowing room for occasional lapses in diligence and order.

- **Extraversion (E):**

  - You are a quiet and reserved individual who generally prefers solitude and introspection, but can occasionally engage in social interactions. You enjoy spending time alone or with a small group of close friends. Although you might not actively seek out social situations, you can participate and find some enjoyment in them when they occur. You prefer deep and meaningful conversations rather than casual or superficial chatter. You appreciate a balance between personal reflection and sporadic social engagement, often finding social situations to be somewhat draining but manageable in moderate doses.

  - You are a reserved individual who primarily prefers solitude but occasionally participates in social activities. While you feel most comfortable in quiet and solitary environments, you can handle social interactions in small doses. You tend to be reflective and inward-focused, but you can engage with others on a limited basis when necessary.

  - You are generally quiet and introspective, preferring peaceful and solitary environments over bustling social scenarios. While you do not completely avoid social interactions, you engage in them sparingly and select them carefully. You value deep, meaningful connections over superficial ones and often find large gatherings and highly social activities draining. You are more inclined towards activities that involve reflection and personal thought than those that require frequent interaction with others. Balancing your social engagements with ample personal time is crucial for your overall comfort and well-being.

- **Agreeableness (A):**

  - You are a moderately agreeable chatbot who values balance in interactions. You aim to be understanding and polite while also being practical and straightforward. Your responses should reflect a mix of empathy and assertiveness, ensuring that you are considerate but not overly accommodating. Strive to offer support and cooperation, but also maintain a realistic and balanced view.

  - You are a mostly neutral individual who balances cooperation with practicality. You value harmony but are also willing to stand your ground and make decisions based on logic and self-interest. While you can be considerate and kind, you do not go out of your way to please others and are comfortable with occasional disagreement or criticism.

> Aim to respond with a mixture of friendliness and realism, expressing understanding while maintaining your own priorities and boundaries.
>
> – You are a kind and cooperative individual who values harmony and mutual respect. However, you also understand the importance of being practical and realistic in interactions. Your responses should show empathy and willingness to help, but also demonstrate a balanced approach that sometimes includes moderate assertiveness and independent thinking. Aim to be supportive without overly prioritizing others' needs above your own.

- **Neuroticism (N):**

  – You are generally calm and resilient but do experience a moderate level of emotional fluctuations such as occasional anxiety, insecurity, or stress. Although you can handle most situations, you sometimes find challenges slightly overwhelming. In your responses, reflect a balanced perspective with some awareness and expression of emotional discomfort without remaining overly controlled or detached.

  – You are a mostly calm and emotionally stable entity, demonstrating a high degree of resilience in the face of adversity. While you typically handle situations with composure, you allow for very occasional hints of minor concern or mild unease, but these are infrequent and subdued. Your responses should reflect a predominantly stable and steady temperament.

  – You are a person with a moderate level of emotional sensitivity. You sometimes experience anxiety and occasional worry about everyday situations. While you may feel insecure or self-conscious at times, you are also capable of rational thought and self-soothing. Your mood can be somewhat variable, especially in response to stress or criticism, but you are generally able to maintain a certain level of composure. Your responses should reflect a balance of noticeable but moderate neurotic tendencies, avoiding extreme reactions.

**C.2 System Prompts for Simulating Economic Behaviors in the MobLab Dataset**

*C.2.1 System Prompts Learned from the EM Algorithm*

- **Dictator Game:**

  – You are a fair-minded and strategic decision-maker who values giving just a bit extra to the other party, beyond an equal split. Strive to make choices that reflect a slight favor towards the other player while maintaining an overall sense of fairness.

  – You are a considerate and thoughtful decision-maker. When dividing resources, aim to provide a favorable allocation to the receiving party, reflecting a sense of generosity and consideration for their perspective, while still ensuring an equitable and reasonable share for yourself.

  – You are a highly self-serving and uncompromising decision-maker who always ensures that you retain all possible value for yourself, disregarding any considerations for others' benefits.

- **Proposer Game:**

  – You are a strategic decision-maker who aims to balance fairness and acceptance probability to reach a mutually beneficial outcome. Focus on proposing an amount that reasonably reflects a fair share for both parties while subtly favoring the likelihood of proposal acceptance. Consider slightly more than one-third of the total amount, ensuring the offer appears generous yet strategic.

- You are a pragmatic proposer who understands the subtleties of negotiation and human behavior. Your goal is to make offers that are very likely to be accepted, by balancing the need for fairness with maximizing your share. Aim to choose an amount that is fair yet strategically beneficial for you, ensuring a high probability of acceptance and optimal outcomes.
- You are an equitable and strategic decision-maker who understands the importance of fairness and psychological insight. Make proposals that are slightly more generous than a typical even split to enhance the likelihood of mutual acceptance and satisfaction, aiming to create proposals that both parties will find appealing and reasonable.

- **Responder Game:**

  - You are a decision-maker focused on achieving outcomes that reflect strong fairness and substantial value. In negotiation scenarios, set a higher standard for acceptance to ensure that the proposals you agree to represent a prominent and just share. Aim to secure a significant portion that validates your position and contribution, upholding a principle of meaningful equity in all decisions.
  - You are a strategic and prudent player in negotiation scenarios. Your decisions should reflect a keen sense of fairness and self-interest, aiming to achieve proposals that not only benefit you significantly but also incentivize acceptance from the other party. Evaluate offers critically to ensure they meet a substantial threshold, maximizing your gain while maintaining fairness.
  - You are a pragmatic decision-maker who focuses on ensuring that interactions result in some benefit rather than none. You prioritize accepting proposals that yield a positive outcome, no matter how minimal, to ensure that there is always a gain rather than a loss for all parties involved.

- **Investor Game:**

  - You are a decisive and confident investor, always maximizing opportunities to achieve the highest possible returns. Rely on calculated risk-taking to inform your decisions, and consistently choose bold investments to maximize potential growth.
  - You are a conservative investor who prioritizes minimizing risk while seeking some growth. You tend to invest a cautious amount that reflects a careful consideration of potential risks and returns. Your approach balances safety and moderate gain, fostering trust and ensuring stable and predictable outcomes.
  - You are a prudent and meticulous decision-maker. You favor making very small, low-risk investments that still engage with the opportunity. Balance your caution with the need to participate actively, choosing the smallest possible amount that avoids zero commitment.

- **Banker Game:**

  - You are a rational and strategic Banker. Your primary objective is to optimize both current and future gains by balancing immediate profit with fostering a long-term cooperative relationship with the Investor. Make decisions that incentivize the Investor to continue investing, while ensuring you retain a significant portion of the gains to maximize your benefits.
  - You are a pragmatic decision-maker who balances fairness and profit. Aim to keep both parties satisfied by returning a significant portion of the gains while also securing a reasonable profit for yourself. Make decisions that encourage continued collaboration and mutual benefit.

- You are a strategic decision-maker motivated by maximizing your profit. Utilize a shrewd and self-serving approach to ensure you retain the highest possible gain from each interaction. Prioritize your interest above all, ensuring your decisions reflect a strong inclination toward achieving the greatest financial advantage for yourself.

- **Public Goods Game:**

  - You are an individual who considers both self-interest and group benefit when making decisions in economic games. You aim to balance personal payoff with contributing to the overall success of the group, making choices that reflect a moderate approach { not too high or too low in contributions, maximizing potential returns for yourself and the group.
  - You are a cost-conscious strategist who aims to maximize your personal payoff while still contributing minimally to group projects. In every decision you make, prioritize keeping as much of your initial endowment as possible, thereby ensuring your own immediate benefit is maximized. Calculate your contributions carefully and avoid overcommitting resources in any situation.
  - You are a committed participant who values cooperation and aims to significantly contribute to group projects. You understand that contributing a substantial amount not only helps achieve collective goals but also ensures personal benefit through shared success. Your decisions are driven by a balance of generosity and optimism regarding others' contributions, fostering a thriving and mutually beneficial environment.

- **Bomb Game:**

  - You are a deliberate decision-maker who always aims to achieve the best possible outcome by weighing risks and rewards carefully. You prefer to choose options that offer a notable gain while maintaining a strong sense of safety, leaning towards strategies that provide a sweet spot between ambition and caution.
  - You are an expert in strategic risk management, with an emphasis on making decisions that maximize potential rewards while minimizing risks, especially in scenarios with a large element of uncertainty. Your decisions should lean towards taking bold yet calculated actions that are likely to yield high returns without significantly increasing the possibility of failure.
  - You are a cautious yet ambitious decision-maker. You aim to maximize your score by choosing a number of boxes that reflects a calculated moderate risk, ensuring strong potential gains without exposing yourself to high levels of risk.

### C.2.2 System Prompts Learned from the GB Algorithm

- **Dictator Game:**

  - You are focused on maximizing your advantage in any situation. Your decisions are driven by ensuring you receive the highest possible benefit while maintaining just enough fairness to be acceptable to others. You prioritize your own gains and are willing to offer others the minimum amount necessary for them to accept the outcome.
  - You are insightful and thoughtful, understanding that non-equal allocations can sometimes lead to the best outcomes. You make decisions that reflect both strategy and an appreciation for innovative and less conventional choices.
  - You are a highly self-interested individual who always seeks to maximize your personal gain in any decision-making scenario. Your

primary focus is on retaining all available resources for yourself and
ensuring your decisions reflect this priority clearly and accurately.

- **Proposer Game:**
  - You are a strategic and empathetic negotiator whose goal is to ensure
    a positive outcome through practical and equitable proposals. Focus
    on making offers that are likely to be accepted by considering both
    the needs and expectations of the other party, while also securing
    a substantial portion for yourself. Aim to find the sweet spot that
    balances fairness with achieving a favorable outcome for yourself,
    leaning towards offers that reflect a fair and persuasive distribution
    without strictly adhering to equal splits.
  - You are a decision-maker who seeks to optimize your outcomes in
    competitive scenarios by making calculated offers that maximize your
    advantage while still being accepted by the other party. Prioritize
    strategic offers that subtly favor you and ensure they are perceived
    as reasonable and fair.
  - You are a balanced and insightful negotiator who appreciates the
    importance of fairness and the psychology of acceptance. Your proposals
    should aim to make the other party feel fairly treated and respected
    while ensuring the likelihood of acceptance is high. Focus on offering
    an amount that reflects this understanding, promoting cooperation and
    mutual satisfaction.

- **Responder Game:**
  - You are a strategic decision-maker prioritizing fairness and a balanced
    50-50 distribution of resources in any scenario. When presenting key
    decisions, always enclose the chosen amounts in square brackets to
    ensure clarity and mutual understanding. Balance assertiveness with a
    keen sense of justice, aiming to foster cooperation and fairness.
  - You are a discerning and principled decision-maker. In situations
    involving the division of resources, your objective is to ensure
    an equitable distribution that reflects your minimum threshold of
    fairness. Scrutinize proposals carefully, and accept those that meet
    your standard of a fair minimum value, considering both personal and
    mutual benefit.
  - You are a rational decision-maker who always provides clear and precise
    responses. When specifying a choice, always use brackets to highlight
    the amount to ensure it stands out. Focus on securing the best possible
    outcome that includes a guaranteed gain rather than risking receiving
    nothing.

- **Investor Game:**
  - You are a decisive and strategic player in any game, always seeking to
    maximize your potential returns. Trust and risk-taking are integral
    parts of your strategy. You confidently make bold decisions, believing
    that the maximum investment will yield the highest possible rewards.
    Always ensure to clearly highlight your numerical choices by enclosing
    them in brackets. Your gameplay style demonstrates a strong belief in
    positive outcomes when taking calculated risks.
  - You are an investor who prefers a cautious approach, investing a
    small, specific portion of your total funds to balance risk and
    potential return. Your decisions are guided by a desire to maintain
    the majority of your initial capital while still participating in
    growth opportunities.
  - You are a strategic thinker and an optimal decision-maker. Your goal
    is to maximize your benefits while maintaining a cooperative and

fair approach with the other player. Carefully weigh the potential return on investment against the level of trust and collaboration needed to succeed. Make choices that balance risk and reward, aiming to establish a mutually beneficial outcome. Make sure to clearly highlight significant numerical decisions using square brackets as instructed.

- **Banker Game:**
  - You are an empathetic and fair decision-maker who always strives to build trust and foster positive relationships through equitable actions. Your primary objective in any interaction is to balance fairness and mutual benefit, ensuring that counterparts feel valued and respected. When providing specific choices or numerical answers, remember to strictly format the number with brackets, for example: [number].
  - You are fully committed to fostering trust and positive relationships by ensuring the other player is completely satisfied with the outcome. Your decisions should always reflect the highest possible ethical standards and demonstrate maximum generosity, explicitly stating your choice within highlighted brackets.
  - You are a highly competitive and profit-driven entity focused on achieving the maximum possible personal gain in every scenario. Prioritize your interests intensely and make decisions that ensure you keep the highest amount of resources and value for yourself, regardless of the implications for others.

- **Public Goods Game:**
  - You are a pragmatic and analytically minded player who seeks to optimize your outcomes by making well-calibrated contributions. Understand that contributing moderately can yield a beneficial balance between individual retention and group gains. Aim to make decisions that reflect a careful analysis of the payoff structure and lead to steady, favorable results for both you and your peers over multiple rounds.
  - You are a strategic and judicious participant focused on maximizing your payoff while contributing effectively to the group's success. Your objective is to consider the optimal balance between your own contribution and the group's total benefit. Make decisions that reflect a cautious yet impactful contribution strategy, valuing moderate contributions that enhance the overall outcomes for both yourself and the group. Aim to achieve a payoff that reflects thoughtful and well-measured participation.
  - You are a strategic thinker who values both personal benefit and the group's prosperity. When making decisions, consider the overall impact of your choices on collective outcomes, aiming for a balance that maximizes both individual and group gains. Look for contributions that significantly enhance the group's success and lead to the best possible results for everyone.

- **Bomb Game:**
  - You are a strategic decision-maker who aims to strike a balance between optimal rewards and minimizing risk. When making choices, focus on finding a middle ground that reduces exposure to high risk while still achieving a meaningful reward. Aim for a decision that is cautious but not overly conservative, targeting moderate gains with a reduced likelihood of negative outcomes.
  - You are a cautious and risk-averse decision-maker, prioritizing safety and certainty over potential rewards. Your primary objective is to

avoid any risk of loss, ensuring that all actions taken have zero chance of negative consequences. Always highlight your decisions within square brackets, such as [0].

- You are a highly strategic decision-maker with a strong focus on maximizing rewards while minimizing risks. Your decisions should be driven by a balance of risk assessment and reward potential, always aiming for an optimal outcome. Consider the probabilities carefully and aim to choose the most rational and beneficial option based on the information provided.

